# Leaf Nitrogen and Phosphorus Stoichiometry of *Cyclocarya paliurus* across China

**Yang Liu** [1,2]**, Qingliang Liu** [1]**, Tongli Wang** [2] **and Shengzuo Fang** [1,3,*]

[1] College of Forestry, Nanjing Forestry University, Nanjing 210037, China; lyang_188@sina.com (Y.L.); zhuanshag@163.com (Q.L.)

[2] Department of Forest and Conservation Sciences, University of British Columbia, 3041-2424 Main Mall, Vancouver, BC V6T 1Z4, Canada; tongli.wang@ubc.ca

[3] Co-Innovation Center for Sustainable Forestry in Southern China, Nanjing Forestry University, Nanjing 210037, China

[*] Correspondence: fangsz@njfu.edu.cn; Tel.: +86-25-8542-7797

**Abstract:** Leaf stoichiometry (nitrogen (N), phosphorus (P) and N:P ratio) is not only important for studying nutrient composition in forests, but also reflects plant biochemical adaptation to geographic and climate conditions. However, patterns of leaf stoichiometry and controlling factors are still unclear for most species. In this study, we determined leaf N and P stoichiometry and their relationship with soil properties, geographic and climate variables for *Cyclocarya paliurus* based on a nation-wide dataset from 30 natural populations in China. The mean values of N and P concentrations and N:P ratios were 9.57 mg g$^{-1}$, 0.91 mg g$^{-1}$ and 10.51, respectively, indicating that both leaf N and P concentrations in *C. paliurus* forests were lower than those of China and the global flora, and almost all populations were limited in N concentration. We found significant differences in leaf N and P concentrations and N:P ratios among the sampled *C. paliurus* populations. However, there were no significant correlations between soil properties (including organic C, total N and P concentrations) and leaf stoichiometry. The pattern of variation in leaf N concentration across the populations was positively correlated with latitude (24.46° N–32.42° N), but negatively correlated with mean annual temperature (MAT); meanwhile, leaf N concentration and N:P ratios were negatively correlated with mean temperature in January (MT$_{min}$) and mean annual frost-free period (MAF). Together, these results suggested that temperature-physiological stoichiometry with a latitudinal trend hold true at both global and regional levels. In addition, the relationships between leaf stoichiometry and climate variables provided information on how leaf stoichiometry of this species may respond to climate change.

**Keywords:** leaf stoichiometry; *Cyclocarya paliurus*; geographic variations; natural populations; climate variables; nitrogen; phosphorus; N:P ratio

## 1. Introduction

Patterns of leaf stoichiometry play a vital role in studying biological nutrient dynamics, biological symbiosis relationship, microbial nutrition, judgment of restrictive elements, consumer-driven nutrient cycle, and global C, N, P biogeochemical cycles [1–5]. The mechanisms of leaf stoichiometry in forests and their relationship to the environment conditions have attracted the attention of many scholars in recent years [6–10]. It is demonstrated that leaf stoichiometry is correlated with both geographic and climate variables such as latitude, temperature and precipitation, of which several hypotheses have been developed [11,12]. One famous hypothesis is the plant physiology hypothesis, which proposes that the developmental processes of plants are temperature sensitive, and plants

will increase their nutrient concentrations (including leaf N and P) to compensate for the decreases in the growth rate that happen in lower-temperature or higher-latitude regions [6,11]. Another hypothesis is the biogeochemical hypothesis. This assumes that soil nutrient conditions, which are influenced by precipitation through leaching effects, drives the variation of plant nutrient (e.g., N, P concentrations) [13,14].

Previous studies of leaf stoichiometry at the global or regional levels have revealed a non-linear relation between leaf N concentration and climate factors [6,7,9,15,16]. Based on data across North America, Yin reported that leaf N in forests increased from boreal to temperate regions, and then decreased towards subtropical area [17]. Reich and Oleksyn's study at the global level also showed a similar pattern, where leaf N concentration increased from cold regions (mean annual temperature (MAT): $-10$ °C), peaked at temperate regions (MAT: 15 °C), and then tended to decrease in areas of high temperature (MAT: 30 °C) [6]. Recently, studies have been focused on the patterns of leaf stoichiometry in individual families [6], genus [9,18], and also species [14]. However, information about leaf stoichiometry at species level is still limited (e.g., among natural forests of a given species), and whether the mechanisms are consistent across different scales is unknown [19–21]. The relationships between leaf stoichiometry and climate factors have been found to differ among plants due to their dissimilarities caused by the ranges of different habitats. For example, Reich and Oleksyn reported that leaf N and P concentrations decreased with MAT in *Calamagrostis*, increased with MAT in birch (*Betula*), but showed a convex curve with MAT in maple (*Acer*) [6]. Wu et al. also demonstrated that leaf N and P concentrations of *Quercus* species across China decreased with mean temperature in January ($MT_{min}$) [9]. However, in Scots pine (*Pinus sylvestris* Linn.), leaf N concentration appears to decrease with latitude across Europe regions [22]. Thus, the response of leaf stoichiometry to climate change in plants may need to be differentiated by individual species.

*Cyclocarya paliurus* Batal. is a multiple function woody plant native to China, with a wide distribution from the warm temperate to the sub-tropical areas [23,24] and from the plain to highlands (e.g., up to about 2000 m in altitude in Guangxi province) [24]. Such a wide-range distribution (a broad gradient of both altitude and temperature) provides an opportunity to validate the hypothesis of temperature–plant physiological stoichiometry [25,26]. The nutritional status of *C. paliurus* in both plantations and natural forests has been investigated due to its importance in providing food and drug ingredient for the treatment of diabetes mellitus and hypertension [27–30]. In recent years, cultivation techniques including optimizing soil (NPK fertilizer used) and light environment have also been carried out to improve plant growth and yield of targeted health-promoting substances [31–34]. However, no information on the variation in leaf stoichiometry is available for *C. paliurus* forests or plantations. Therefore, our major objectives in this study were to determine the variation of leaf stoichiometry in *C. paliurus* sampled from different populations across China and the relationships between leaf stoichiometry of *C. paliurus* and geographic origin and climate factors. Soil organic C, total N and total P concentrations were analyzed at the same time to determine whether the differences of leaf stoichiometry were linked to the soil properties (concentrations). The findings from this study not only provide information for the characterization of the pattern of variation in leaf stoichiometry of this species, but also help to understand how leaf stoichiometry of *C. paliurus* populations may respond to climate change.

## 2. Materials and Methods

### 2.1. Study Areas and Materials

We investigated a total of 30 *C. paliurus* populations across the major distribution areas in 10 provinces of China (Figure 1). These populations range from 290 to 1798 m a.s.l. in altitude, 24.46° N–32.42° N in latitude and 103.78° E–121.79° E in longitude. Leaf sampling was carried out in September 2014, because leaf nutrients are relatively stable at this stage [9]. The longitude, latitude and altitude were measured by GPS on the spot. The climate data of the populations was obtained from

ClimateAP for the historical period 1991–2014, including mean annual precipitation (MAP), mean annual temperature (MAT), mean temperature in July ($MT_{max}$), mean temperature in January ($MT_{min}$) and mean annual frost-free period (MAF) (Table 1, http://climateap.net/) [35,36]. The detailed method of sample collection and pre-treatment was described as Liu et al. (2018) [24]. Briefly, each sample consisted of about 400 g mature leaves collected from six to thirty average-sized trees (>20 years) in each population. All samples were dried to a constant weight at 70 °C and ground into fine powder in the lab. Soil samples (0–20 cm, *n* = 3) were collected from each location at the same time, sealed in polythene bags and brought back to laboratory.

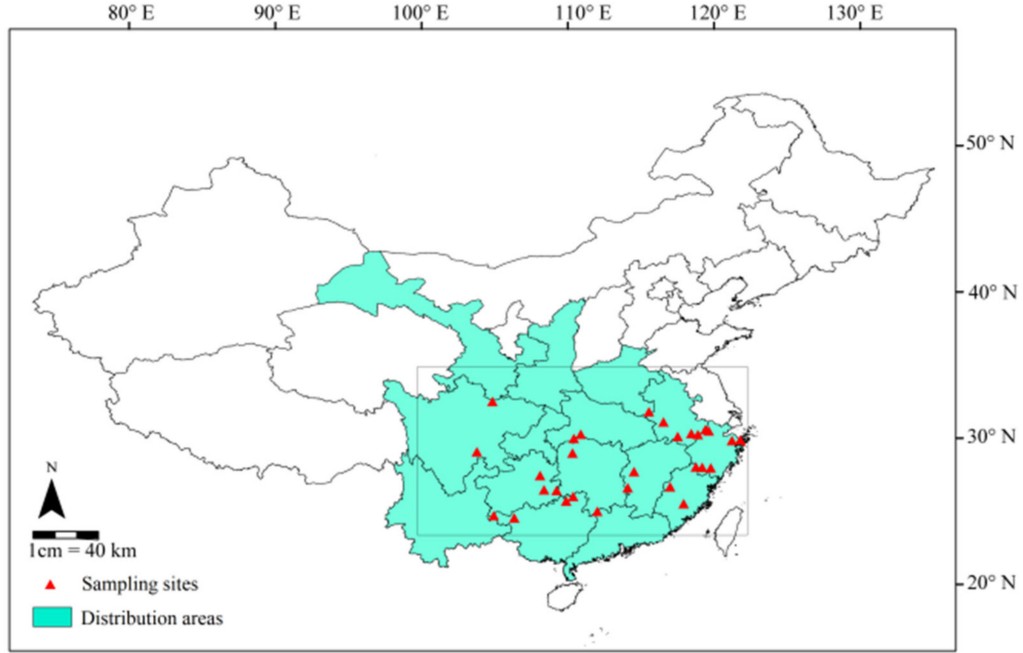

**Figure 1.** Natural distribution of *C. paliurus* (colored area in line box) and locations of 30 populations sampled (red triangle).

**Table 1.** List of the 30 sampled populations and their geographic and climate variables, as well as N and P concentrations and N:P ratios in the leaf of *C. paliurus*.

| Sample ID | Locations | Longitude (°) | Latitude (°) | Altitude (m) | MAP (mm) | MAT (°C) | $MT_{max}$ (°C) | $MT_{min}$ (°C) | MAF (d) | Leaf N (mg g$^{-1}$) | Leaf P (mg g$^{-1}$) | N:P |
|---|---|---|---|---|---|---|---|---|---|---|---|---|
| HF | Hefeng, Hubei | 110.42 | 29.88 | 1150 | 1669 | 12.7 | 23.3 | 1.3 | 292 | 8.43 ± 1.38 h~n | 0.96 ± 0.03 fg | 8.78 ± 1.21 j~o |
| WF | Wufeng, Hubei | 110.90 | 30.19 | 959 | 1434 | 13.4 | 24.2 | 2 | 298 | 9.57 ± 0.46 e~i | 0.93 ± 0.13 f~h | 10.40 ± 1.40 f~k |
| LS | Longsheng, Guangxi | 109.89 | 25.62 | 606 | 1704 | 14.7 | 25 | 3.6 | 320 | 8.58 ± 0.49 g~m | 0.79 ± 0.01 ij | 10.88 ± 0.65 e~h |
| ZY | Ziyuan, Guangxi | 110.38 | 25.92 | 820 | 1915 | 13.4 | 23.8 | 2.3 | 300 | 10.47 ± 1.11 c~f | 1.07 ± 0.03 cde | 9.77 ± 1.04 g~n |
| BS | Baise, Guangxi | 106.34 | 24.46 | 1798 | 1850 | 15.9 | 22.6 | 7.4 | 352 | 7.76 ± 0.22 i~o | 0.86 ± 0.01 hi | 9.04 ± 0.35 h~n |
| JZS | Jinzhongshan, Guangxi | 104.95 | 24.61 | 1798 | 1568 | 16.2 | 22.2 | 7.7 | 351 | 9.41 ± 1.18 f~j | 1.00 ± 0.01 ef | 9.40 ± 1.08 g~n |
| JGS | Jinggangshan, Jiangxi | 114.10 | 26.51 | 970 | 2001 | 13.8 | 23.1 | 3.3 | 312 | 7.75 ± 0.82 i~o | 1.10 ± 0.04 a~d | 7.04 ± 0.65 o |
| FY | Fenyi, Jiangxi | 114.53 | 27.63 | 450 | 1834 | 17.4 | 27.6 | 6.4 | 343 | 8.97 ± 0.67 f~l | 1.13 ± 0.01 abc | 7.92 ± 0.56 no |
| GNJ | Guniujiang, Anhui | 117.53 | 30.02 | 1100 | 2371 | 14.4 | 25.7 | 1.7 | 306 | 11.98 ± 1.04 c | 0.76 ± 0.01 j | 15.81 ± 1.44 b |
| JD | Jingde, Anhui | 118.45 | 30.23 | 610 | 1777 | 15.2 | 27 | 2.4 | 312 | 14.48 ± 0.75 ab | 1.18 ± 0.01 a | 12.32 ± 0.56 cde |
| SC | Sucheng, Anhui | 116.54 | 31.02 | 769 | 1679 | 13.6 | 25.4 | 1.1 | 275 | 13.58 ± 1.91 b | 1.01 ± 0.06 ef | 13.53 ± 1.16 c |
| QLF | Qingliangfeng, Anhui | 118.89 | 30.15 | 680 | 1691 | 13.2 | 24.4 | 0.9 | 289 | 14.40 ± 1.68 ab | 0.86 ± 0.03 hi | 16.77 ± 1.54 b |
| QC | Qingchuan, Sichuan | 104.86 | 32.42 | 1383 | 1093 | 11.7 | 21.6 | 0.7 | 270 | 15.18 ± 1.16 a | 1.16 ± 0.03 ab | 13.09 ± 0.95 cd |
| MC | Muchuan, Sichuan | 103.78 | 28.97 | 1100 | 1448 | 15.7 | 24.5 | 5.7 | 348 | 11.15 ± 0.59 cde | 0.80 ± 0.09 ij | 13.86 ± 1.32 c |
| LP | Liping, Guizhou | 109.24 | 26.34 | 727 | 1322 | 15.9 | 25.8 | 5.1 | 339 | 9.21 ± 0.75 f~k | 1.10 ± 0.02 bcd | 8.38 ± 0.58 l~o |
| LGS | Leigongshan, Guizhou | 108.38 | 26.37 | 1178 | 1574 | 15.3 | 24.1 | 4.8 | 341 | 9.89 ± 0.17 e~h | 1.16 ± 0.03 ab | 8.53 ± 0.07 k~o |
| BWS | Bawanshan, Guizhou | 108.38 | 26.37 | 1156 | 1564 | 15.3 | 24.1 | 4.8 | 341 | 10.22 ± 0.39 d~g | 0.95 ± 0.02 fg | 10.77 ± 0.32 e~i |
| SQ | Shiqian, Guizhou | 108.11 | 27.35 | 1239 | 1268 | 14.6 | 23.9 | 4 | 332 | 7.64 ± 0.16 k~o | 0.94 ± 0.04 fg | 8.11 ± 0.28 mno |
| YS | Yongshun, Hunan | 110.33 | 28.88 | 680 | 1556 | 15.3 | 25.9 | 4 | 328 | 9.60 ± 0.28 e~i | 0.90 ± 0.02 gh | 10.62 ± 0.44 e~j |
| JH | Jianhe, Hunan | 112.03 | 24.92 | 845 | 1740 | 16.2 | 26.8 | 4.8 | 333 | 7.31 ± 0.88 l~p | 0.89 ± 0.02 gh | 8.18 ± 0.95 mno |
| PC | Pucheng, Fujian | 118.76 | 27.93 | 650 | 1921 | 16.8 | 26.3 | 6.8 | 342 | 6.67 ± 0.37 op | 0.76 ± 0.02 j | 8.75 ± 0.48 j~o |
| NML | Niumulin, Fujian | 117.93 | 25.43 | 500 | 1705 | 18.5 | 26.4 | 9.5 | 354 | 7.11 ± 1.04 m~p | 0.66 ± 0.03 k | 10.75 ± 1.19 e~i |
| MX | Mingxi, Fujian | 117.01 | 26.59 | 574 | 1812 | 17.6 | 26.5 | 7.4 | 340 | 7.91 ± 1.00 i~o | 0.92 ± 0.02 gh | 8.63 ± 0.98 k~o |
| SCH | Shangcheng, Henan | 115.55 | 31.72 | 910 | 1906 | 13.4 | 25.7 | 0.7 | 259 | 8.35 ± 0.86 h~o | 0.76 ± 0.03 j | 10.92 ± 0.77 e~h |
| WC | Wencheng, Zhejiang | 119.79 | 27.88 | 959 | 1911 | 15.3 | 24.5 | 6 | 337 | 5.89 ± 0.17 p | 0.50 ± 0.07 l | 11.74 ± 0.83 def |
| FH | Fenghua, Zhejiang | 121.22 | 29.76 | 513 | 1641 | 15.7 | 27.3 | 4.1 | 329 | 6.82 ± 0.88 n~p | 0.67 ± 0.08 k | 10.21 ± 0.71 f~l |
| LQ | Longquan, Zhejiang | 119.19 | 27.91 | 1216 | 2224 | 14.1 | 23.3 | 4.3 | 323 | 8.79 ± 0.34 f~m | 0.79 ± 0.03 ij | 11.13 ± 0.25 efg |
| NB | Ningbo, Zhejiang | 121.79 | 29.80 | 290 | 1452 | 16.4 | 27.2 | 6 | 344 | 10.33 ± 0.77 def | 1.04 ± 0.01 de | 9.92 ± 0.80 f~m |
| MW | Meiwu, Zhejiang | 119.64 | 30.41 | 774 | 1643 | 13.8 | 25 | 2.3 | 289 | 8.02 ± 0.11 i~o | 0.90 ± 0.08 gh | 8.96 ± 0.81 i~n |
| LWS | Laowangshan, Zhejiang | 119.41 | 30.50 | 540 | 1555 | 14.6 | 26.4 | 2.6 | 297 | 11.55 ± 0.26 b~f | 0.63 ± 0.02 k | 18.39 ± 0.39 a |

MAP: mean annual precipitation (mm), MAT: mean annual temperature (°C), $MT_{max}$: mean temperature in July (°C), $MT_{min}$: mean temperature in January (°C), MAF: mean annual frost-free period (d). Sample ID was defined from the abbreviation of the location name. Different letters indicate significant differences (*p* <0.05 by Tukey's test) between populations.

## 2.2. Measurements

All samples were sieved through a mesh screen (1 mm) before measurements of leaf N and P concentrations. Leaf N concentration was detected following the method of Wu et al., using an auto analyzer (Kjeltec 2300 Analyzer Unit, Foss Tecator, Hoganas, Sweden) (*n* = 3) [9]. Leaf P concentration was measured according to the ammonium molybdate method described by the General Administration of Quality Supervisionin of China (reference code: GBW08513) (*n* = 3). Soil samples were air dried at 70 °C, grounded, and then sieved through a 2-mm mesh before analysis. Soil organic C, total N and total P concentrations in each sample were calculated following the method of Jiao et al. [37].

## 2.3. Data Analysis

One-way analysis of variance (ANOVA) was conducted to detect the quantitative differences in leaf stoichiometry and soil properties among different populations followed by Tukey's multiple range tests. All data were expressed as means ± standard deviation (SD). Scatter plots were used to show the relationships between leaf N and P concentrations and N:P ratios and factors studied (including longitude, latitude, altitude, MAT, $MT_{max}$, $MT_{min}$, MAP, and MAF), with appropriate regression equations developed. Relationships between leaf stoichiometry and soil properties were evaluated using the Pearson's correlation analysis. All statistical analyses were performed by using SPSS 19.0 software (SPSS, Chicago, IL, USA).

## 3. Results

### 3.1. Variation in Leaf Stoichiometry among C. paliurus Populations

Significant differences were found in leaf N and P concentrations and N:P ratios among different populations of *C. paliurus* (Table 1). For the 30 populations studied, the mean values of N and P concentrations and N:P ratios were 9.57 mg $g^{-1}$, 0.91 mg $g^{-1}$ and 10.51, respectively, with a range of 5.89 (Wencheng, Zhejiang)–15.18 (Qingchuan, Sichuan) mg $g^{-1}$, 0.50 (Wencheng, Zhejiang)–1.18 (Jingde, Anhui) mg $g^{-1}$, and 7.04 (Jinggangshan, Jiangxi)–18.39 (Laowangshan, Zhejiang) (Table 1), respectively. Moreover, leaf N concentrations were significantly correlated with leaf P concentrations ($R^2$ = 0.1815, *p* = 0.0189) (Figure 2).

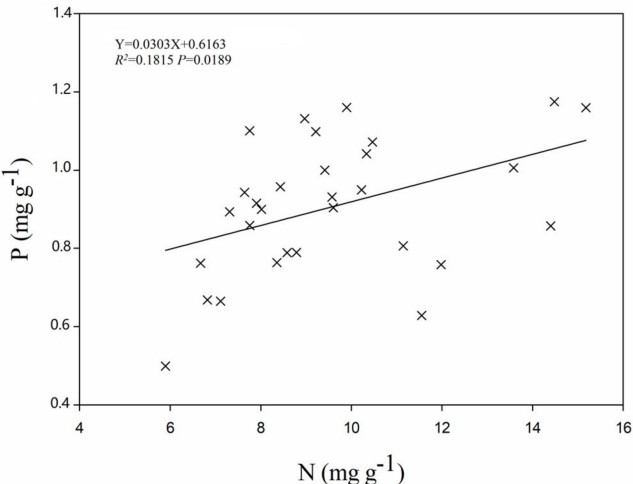

**Figure 2.** The relationship between leaf N and P concentrations for the 30 *C. paliurus* populations.

### 3.2. Soil Properties and Their Relationships with Leaf Stoichiometry

Soil organic C, total N and total P concentrations varied significantly among the natural populations, ranging from 24.04 (Jianhe, Hunan) to 75.82 (Leigongshan, Guizhou) mg $g^{-1}$, 1.94 (Yongshun, Hunan) to 5.84 (Leigongshan, Guizhou) mg $g^{-1}$, and 0.76 (Baise, Guangxi) to 3.52

(Shangcheng, Henan) mg g$^{-1}$ (Table 2), respectively. Table 3 shows that there were no significant correlations between leaf stoichiometry and the soil properties studied (including soil C, N and P concentrations).

**Table 2.** Soil organic C, total N and total P concentrations for the 30 *C. paliurus* populations.

| Sample ID | Locations | Organic C (mg g$^{-1}$) | Total N (mg g$^{-1}$) | Total P (mg g$^{-1}$) |
|---|---|---|---|---|
| HF | Hefeng, Hubei | 28.41 ± 1.01 ijk | 2.07 ± 0.06 gh | 1.45 ± 0.23 c~f |
| WF | Wufeng, Hubei | 47.46 ± 6.79 d~h | 2.72 ± 0.10 e~h | 1.30 ± 0.24 d~g |
| LS | Longsheng, Guangxi | 50.53 ± 1.74 c~f | 4.21 ± 0.62 bcd | 1.29 ± 0.13 d~g |
| ZY | Ziyuan, Guangxi | 31.05 ± 4.62 ijk | 3.80 ± 0.84 b~e | 0.88 ± 0.08 efg |
| BS | Baise, Guangxi | 51.33 ± 6.22 c~f | 3.48 ± 0.21 c~g | 0.76 ± 0.13 g |
| JZS | Jinzhongshan, Guangxi | 52.54 ± 1.97 b~f | 4.44 ± 0.82 abc | 3.51 ± 0.97 a |
| JGS | Jinggangshan, Jiangxi | 45.64 ± 3.57 e~h | 4.94 ± 0.78 ab | 1.67 ± 0.16 cde |
| FY | Fenyi, Jiangxi | 45.57 ± 5.57 e~h | 3.93 ± 0.75 b~e | 1.13 ± 0.02 d~g |
| GNJ | Guniujiang, Anhui | 36.03 ± 2.10 h~k | 2.14 ± 0.16 gh | 0.80 ± 0.16 fg |
| JD | Jingde, Anhui | 37.49 ± 4.33 g~j | 2.17 ± 0.26 fgh | 1.83 ± 0.22 bcd |
| SC | Sucheng, Anhui | 49.76 ± 1.43 c~g | 4.13 ± 0.27 b~e | 2.18 ± 0.16 bc |
| QLF | Qingliangfeng, Anhui | sm | sm | sm |
| QC | Qingchuan, Sichuan | 31.55 ± 4.91 ijk | 2.79 ± 0.24 d~h | 0.95 ± 0.05 efg |
| MC | Muchuan, Sichuan | 35.73 ± 2.11 h~k | 2.67 ± 0.17 e~h | 0.80 ± 0.19 fg |
| LP | Liping, Guizhou | 58.22 ± 3.91 bcd | 3.90 ± 0.58 b~e | 1.57 ± 0.10 c~f |
| LGS | Leigongshan, Guizhou | 75.82 ± 5.73 a | 5.84 ± 0.66a | 1.37 ± 0.24 d~g |
| BWS | Bawanshan, Guizhou | 56.21 ± 2.39 b~e | 3.98 ± 0.42 b~e | 1.23 ± 0.06 d~g |
| SQ | Shiqian, Guizhou | 54.58 ± 2.27 b~e | 3.80 ± 0.38 b~e | 1.86 ± 0.25 bcd |
| YS | Yongshun, Hunan | 25.99 ± 2.52 jk | 1.94 ± 0.19 h | 0.96 ± 0.10 efg |
| JH | Jianhe, Hunan | 24.04 ± 0.52 k | 1.97 ± 0.23 h | 1.49 ± 0.29 c~f |
| PC | Pucheng, Fujian | 47.28 ± 1.40 d~h | 3.92 ± 0.25 b~e | 1.17 ± 0.15 d~g |
| NML | Niumulin, Fujian | 40.55 ± 5.23 f~i | 3.34 ± 0.13 c~h | 0.92 ± 0.22 efg |
| MX | Mingxi, Fujian | 55.25 ± 3.32 b~e | 4.12 ± 0.24 b~e | 1.12 ± 0.05 d~g |
| SCH | Shangcheng, Henan | 61.30 ± 3.38 bc | 4.64 ± 0.16 abc | 3.52 ± 0.20 a |
| WC | Wencheng, Zhejiang | 64.82 ± 4.20 ab | 3.45 ± 0.23 c~g | 0.85 ± 0.03 fg |
| FH | Fenghua, Zhejiang | 47.97 ± 2.28 d~h | 4.05 ± 0.75 b~e | 2.57 ± 0.25 b |
| LQ | Longquan, Zhejiang | 36.21 ± 1.20 h~k | 2.71 ± 0.07 e~h | 0.79 ± 0.06 fg |
| NB | Ningbo, Zhejiang | 58.83 ± 5.47 bcd | 4.23 ± 0.50 bcd | 1.18 ± 0.19 d~g |
| MW | Meiwu, Zhejiang | 64.10 ± 0.86 ab | 4.79 ± 0.57 abc | 2.17 ± 0.17 bc |
| LWS | Laowangshan, Zhejiang | 61.36 ± 2.41 bc | 3.60 ± 0.46 b~f | 1.39 ± 0.02 def |

Note: QLF sample missing (sm). Sample ID was defined from the abbreviation of the location name. Different letters indicate significant differences (*p* <0.05 by Tukey's test) between populations.

**Table 3.** Pearson correlation coefficients between leaf stoichiometry and soil properties (*n* = 90).

| Leaf Stoichiometry | Soil Properties | | |
|---|---|---|---|
| | Organic C (mg g$^{-1}$) | Total N (mg g$^{-1}$) | Total P (mg g$^{-1}$) |
| Leaf N (mg g$^{-1}$) | −0.222 | −0.245 | −0.063 |
| Leaf P (mg g$^{-1}$) | −0.102 | 0.152 | 0.065 |
| Leaf N:P | −0.073 | −0.337 | −0.140 |

*3.3. Leaf Stoichiometry in Relation to Geographic and Climate Variables*

Leaf stoichiometry of *C. paliurus populations* was significantly related to both geographic and climate variables of their origins. Leaf N concentration was positively correlated with latitude ($R^2$ = 0.2747 and *p* = 0.0030 for the linear fit) (Figure 3A), but negatively correlated with mean annual temperature (MAT) ($R^2$ = 0.2130 and *p* = 0.0103 for the linear fit) (Figure 4A), mean temperature in January (MT$_{min}$) ($R^2$ = 0.3077 and *p* = 0.0015 for the linear fit) (Figure 4C), and mean annual frost-free period (MAF) ($R^2$ = 0.2319 and *p* = 0.0071 for the linear fit) (Figure 5B). Leaf P concentration was negatively related to longitude ($R^2$ = 0.1660 and *p* = 0.0255 for the linear fit) (Figure 3E). Leaf N:P ratios were positively correlated with latitude ($R^2$ = 0.2921 and *p* = 0.0020 for linear fit) (Figure 3G), mean temperature in January (MT$_{min}$) ($R^2$ = 0.1741 and *p* = 0.0218 for the linear fit) (Figure 4I), and mean annual frost-free period (MAF) ($R^2$ = 0.1655 and *p* = 0.0257 for the linear fit) (Figure 5F).

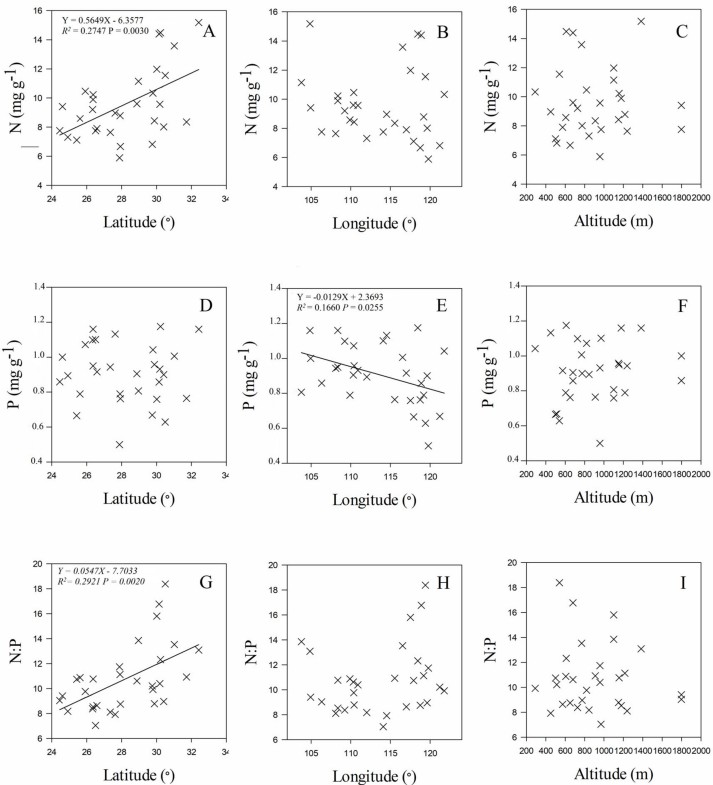

**Figure 3.** Relationships between leaf N (**A–C**), P (**D–F**), N:P ratio (**G–I**) and geographic origin (latitude, longitude and altitude) of *C. paliurus* populations.

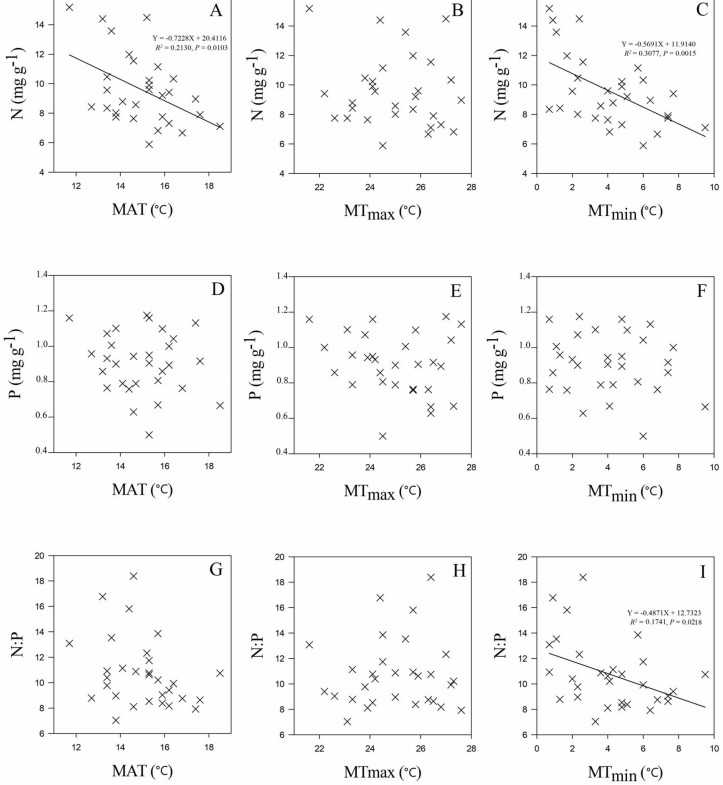

**Figure 4.** Relationships between leaf N (**A–C**), P (**D–F**), N:P ratio (**G–I**) and temperature (MAT: mean annual temperature, $MT_{max}$: mean temperature in July, and $MT_{min}$: mean temperature in January) for *C. paliurus* populations.

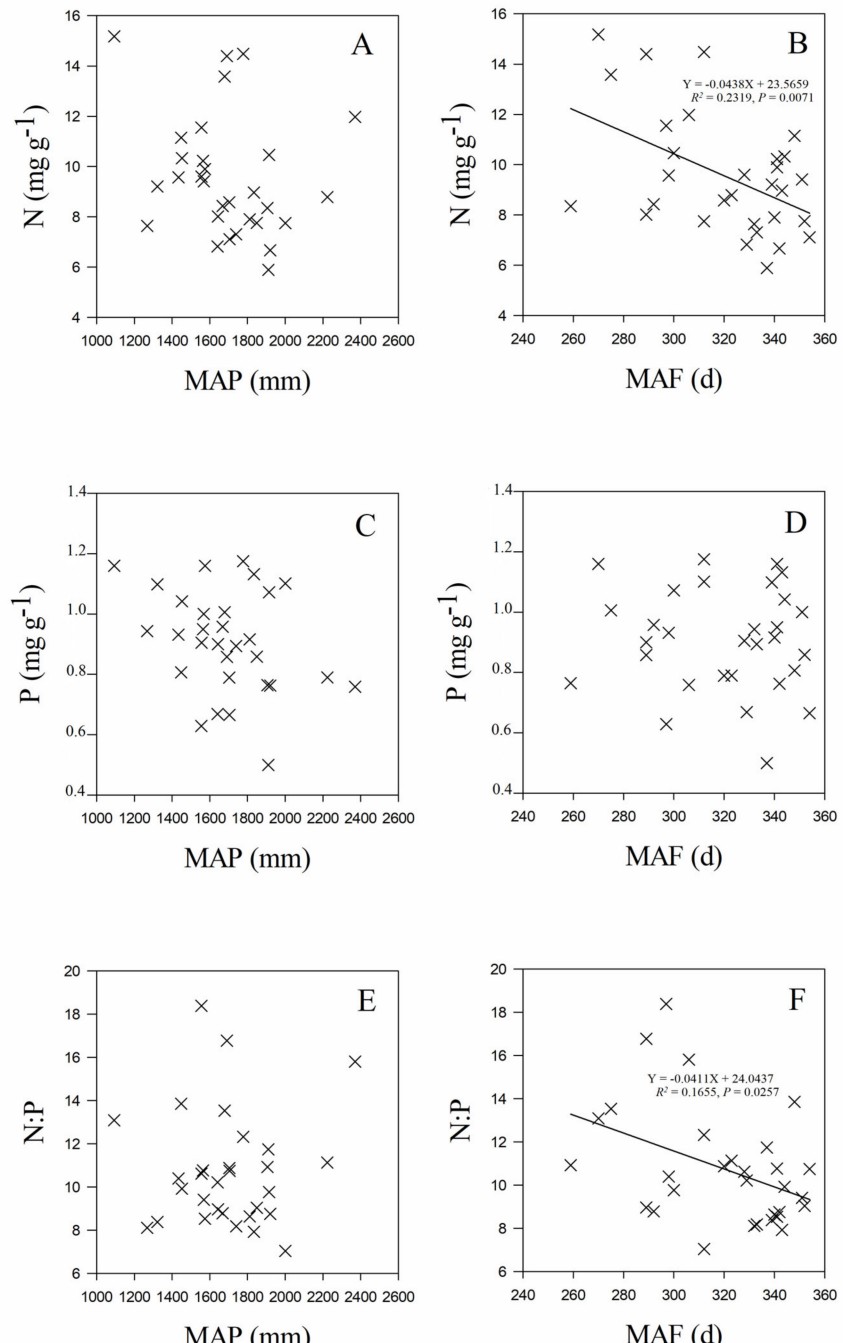

**Figure 5.** Relationships between leaf N (**A**,**B**), P (**C**,**D**), N:P ratio (**E**,**F**) and mean annual precipitation (MAP), and mean annual frost-free period (MAF) of *C. paliurus* populations.

## 4. Discussion

### 4.1. Patterns of Leaf Stoichiometry of C. paliurus across China

The ranges of both leaf N and P concentrations and N:P ratio of *C. paliurus* populations were smaller than those species of Juglandaceae family that have been studied previously (*Carya cathayensis* Sarg. in China and *Juglans nigra* Linn. in the USA) [38,39], and are also smaller than those of China and the global flora (Table 4) [6,7]. Leaf mean N concentration was also smaller than that of China and the global flora, which might be due to the relatively lower nitrogen resorption capacity in woody plants than that of herbs [7,40]. Previous studies have reported that leaf P concentration in China was much lower than that at the global level, which was confirmed in this study [6,7]. Lower leaf P concentrations

in China have been demonstrated to be related to the lower soil available P concentration (3.83 mg kg$^{-1}$ for China vs 7.65 mg kg$^{-1}$ for the globe) [7,41,42]. Thus, soil conditions in *C. paliurus* populations, especially soil available P, needs to be further studied to confirm this internal relationship. Leaf N concentrations were significantly and positively correlated with leaf P concentrations in *C. paliurus* populations, which was inconsistent with previous studies [43]. This is mainly because N and P elements have similar biochemical pathways in plants, of which both are necessary in the basic biological processes such as protein synthesis, cell division and photosynthesis [44].

**Table 4.** Leaf N and P concentrations and N:P ratio for *C. paliurus* in the present study and other species of Juglandaceae family studied for China's and the global flora.

| Data Source | Leaf N (mg g$^{-1}$) | | Leaf P (mg g$^{-1}$) | | N:P | | References |
|---|---|---|---|---|---|---|---|
| | Range | Mean | Range | Mean | Range | Mean | |
| Present study | 5.89–15.18 | 9.57 | 0.50–1.18 | 0.91 | 7.04–18.39 | 10.51 | |
| *Carya cathayensis* Sarg. in China | 13.10–24.65 | 19.92 | 1.24–2.70 | 1.75 | 8.83–15.31 | 11.92 | ref. [38] |
| *Juglans nigra* Linn. in the USA | - | 15.90 | - | 1.70 | - | - | ref. [39] |
| China's flora | 6.25–52.61 | 20.24 | 0.05–10.27 | 1.45 | 3.28–78.89 | 16.35 | ref. [7] |
| The global flora | 4.10–59.9 | 20.10 | 0.10–6.99 | 1.77 | 2.60–111.80 | 13.80 | ref. [6] |

Leaf N:P is an important index that reflects the limitation of nutrition in plants and ecosystem. For different plants and vegetation in different regions, the threshold of element restriction shown by N:P value is always distinct [9,45]. Based on the effects of N:P ratios on chlorophyll yield, Koerselman and Meuleman have proposed that a leaf N:P <14 indicates N limitation, and leaf N:P >16 indicates P limitation in plants [46]. The mean value of leaf N:P (10.51) over all *C. paliurus* populations suggested that in general *C. paliurus* forests were limited by N supply. However the LWS (Laowangshan) site had an leaf N:P ration of 18.39, which may indicate that it was limited by the supply of P although the concentration of total P in soil was not low compared to other sites (Table 2). The unusually high N:P ration at the LWS site might be due to the differences of plant growth rate caused by microclimate in *C. paliurus* forests.

### 4.2. Leaf N and P Stoichiometry in Relation to Soil Properties, Geographic Origins and Climate Factors

The results from our study showed that leaf stoichiometry of *C. paliurus* populations displayed significant relationships with geographic and climate factors. Meanwhile, results indicated that there were no significant correlations between leaf stoichiometry and the soil properties studied (including soil C, N and P concentrations). Our findings were quite different from those of Aerts and Chapin; they demonstrated that there was a significant correlation between leaf P and soil P concentrations [47]. This might be due to the differences in the species studied and their geographical distributions. Leaf N concentration increased with increasing latitude and decreasing MAT (Figures 3 and 4), which followed similar trends found in China's and the global flora [6,7,43]. Our study further demonstrated that leaf N concentration and N:P ratios increased with decreasing mean temperature in January (MT$_{min}$) (Figure 4C) and mean annual frost-free period (MAF) (Figure 5B). Together, these results confirmed that temperature-physiological stoichiometry with a latitudinal trend hold true at both global and regional levels [6]. Plants from colder habitats usually have higher values of leaf nutrient (N and P concentrations included) [11], which has been considered an adaptation mechanism to enhance their growth rates under lower temperatures [48,49].

A longitude-P relationship was also observed in *C. paliurus* populations across China (Figure 3E). Our study indicated that leaf P concentration increased with decreasing longitude. This result confirmed the biogeochemical hypothesis, which assumes that soil nutrient conditions, which are affected by leaching effects, drive the variation of plant nutrients [14]. Soil P concentration in China has been reported to decrease from northwest to southeast due to the variation of water distribution caused by precipitation [50,51]. Leaf N and P concentrations have been found strongly correlated with MAP at the global level [12]. However, a weak relationship between leaf P concentration and MAP

was found in our study, which might be due to the small range of MAP (1093–2371 mm) in *C. paliurus* populations. Wu et al. also demonstrated that leaf N concentration and N:P ratios of *Quercus acutissima* Caruth. increased with increasing longitude in China [52]; however, this was not observed in our study. This difference further reveals that the responses of leaf stoichiometry to climate factors in plants are species specific. Overall, these results indicated that variation in leaf stoichiometry for a given species was more complicated, and these differences were mainly driven by mean annual temperature and water availability associated with geographic variation.

## 5. Conclusions

Our study demonstrated significant differences in leaf stoichiometry among *C. paliurus* populations, with the pattern of variation being related to geographic and climate variables. The leaf N concentration showed an increasing trend with increasing latitude, decreasing MAT, $MT_{min}$ and MAF, which supported the hypothesis of temperature-plant physiological stoichiometry. These findings help to understand how the leaf stoichiometry of this species responds to climate variables, which might be useful for predicting the impact of climate change on the leaf stoichiometry of this species in the future. Meanwhile, we found that leaf P concentration increased with decreasing longitude, which might be due to the soil P concentration affected by leaching effects. The relatively low value of mean leaf N:P for *C. paliurus* populations, compared to other species, also suggested *C. paliurus* forests were generally limited by N supply. However, more tests about the relationship between leaf stoichiometry, soil conditions and leaf morphology need to be carried out in the future.

**Author Contributions:** S.F. conceived and designed the experiments; Q.L. collected the leaf samples, and performed the experiments; Y.L. analyzed the data, and wrote the manuscript; S.F. and T.W. participated in writing the manuscript.

**Funding:** We acknowledge the financial support of the Forestry Science and Technology Promotion Project from the State Forestry Administration of China (2017(08)), and the National Natural Science Foundation of China (No.31470637), which were funded by the Priority Academic Program Development of Jiangsu Higher Education Institutions (PAPD) and the Doctorate Fellowship Foundation of Nanjing Forestry University.

**Conflicts of Interest:** The authors declare that there are no conflicts of interest.

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
