# Peer review of "Leaf Nitrogen and Phosphorus Stoichiometry of Cyclocarya paliurus across China"

_forests, doi:10.3390/f9120771_

Round 1

Reviewer 1 Report

This manuscript describes a study of N and P stoichiometry of a single species across China. This is not the first study of such a geographical variation and is largely confirming what other studies have shown. The novelty in the study is that it shows that patterns observed when many species are pooled are also valid at the individual species level. He data assembled could be a valuable addition in other global studies.

Specific comments

1.     Line 55. What is meant by “constant” mechanism? This should be expressed more clearly.

2.     Lines 173-176. This last sentence is not clear and it is not obvious that OLS and LWS refer to specific stands from Table 1.

Author Response

Thank you very much. We sincerely thank the reviewers for their comments on our manuscript. We have carefully revised the manuscript according to the comments of reviewers.

Reviewer 2 Report

This is an upright study on tending the N & P nutrient concentrations at geographic scales and in relation to climatic variables. However, it provides no information on the most critical contributor to concentrations, the soil (I know this study is not relating to soil, but a bench scale info on the same is required even if derived/synthesized from available literature). This research should be published once the review is addressed.

Line 15. Individual is redundant here

Line 15-16. Why use N and P acronyms again?

Line 17. What is national-wide (I thought nation-wide)?

Line 18. How did you calculate N:P ratio (using your data it comes to 10.51)?

Line 20. Limited in N concentration

Abstract: Briefly mention in the abstract, what range latitude was studied. So far it looks like the use of words stoichiometry especially physiological stoichiometry is fancy

Line 83. Use of “across” instead of “throughout” is suggested

Line 91. Change generally with briefly

Line 93. The degree symbol is unusual

Line 97-98. What author mean “with an autoanalyzer was used”? just say “using an autoanalyzer:

In materials and methods or results, no analysis on soil is given which would be very helpful in determining if the differences were more closely linked to the soil properties (concentrations). A strong argument as to why soil compositions and chemistry of N and P were irrelevant is required???

Fig 2. Provide a regression equation

Fig 3. No substantial differences between linear and quadratic are shown, therefore, later is not needed

Line 157. Might be

Line 160. Be related

Line 171-172. Revise and correct this sentence. Use a”” instead of “an”. Also, add s to indicate

line 183. Correct space

line 188. The use of space around “-“ for expressing relationships is inconsistent throughout the manuscript, need consistency or follow journal formatting guidelines

line 201. Use “characteristic of” instead of “induced from (a weird use of English)”

line 204. Earlier, authors used “climate variable” now climatic variable – so inconsistent

line 208. Can it be used as a proxy?

Author Response

(The authors gave the same response as above.)
